# Rethinking Data Heterogeneity in Federated Learning: Introducing a New Notion and Standard Benchmarks

## Abstract

Though successful, federated learning presents new challenges for machine learning, especially when the issue of data heterogeneity, also known as Non-IID data, arises. To cope with the statistical heterogeneity, previous works incorporated a proximal term in local optimization or modified the model aggregation scheme at the server side or advocated clustered federated learning approaches where the central server groups agent population into clusters with jointly trainable data distributions to take the advantage of a certain level of personalization. While effective, they lack a deep elaboration on what kind of data heterogeneity and how the data heterogeneity impacts the accuracy performance of the participating clients. In contrast to many of the prior federated learning approaches, we demonstrate not only the issue of data heterogeneity in current setups is not necessarily a problem but also in fact it can be beneficial for the FL participants. Our observations are intuitive: (1) Dissimilar labels of clients (label skew) are not necessarily considered data heterogeneity, and (2) the principal angle between the agents' data subspaces spanned by their corresponding principal vectors of data is a better estimate of the data heterogeneity. Our code is available at https://github.com/anonresearcher1/alg-neuripsw22.

## 1 Introduction

Deep learning has emerged as a fast development technique in computer vision, natural language processing, and conversational AI. Though successful, the efficacy of machine learning and deep learning algorithms relies on large quantities of data. However, in areas such as health care, the data may be distributed across numerous hospitals or data centers and cannot be accessed by a central server or cloud due to privacy constraints. For instance, hospitals may have only a few of images of a particular cancer and must be kept private. A lot of works on privacy preserving data management and data mining [1] in a centralized setting have been proposed so far, however, they cannot tackle the cases of distributed databases. Driven by such realistic requirements, in order to conduct data mining/machine learning, it is necessary to exploit data from such distributed databases while preserving data privacy. Federated learning (FL) [2] rise to this challenge due to its ability to collectively train neural networks while preserving data privacy. One of the standard FL algorithms is FedAvg [2]. In each round of FedAvg, clients train models with their local datasets independently, and then the server aggregates the locally trained models, and finally, the parameters of local models are averaged element-wise by the server to obtain a shared *Global* model [3]. FL however introduces distinct issues not present in classical distributed learning. One of the challenges that currently confine the applicability of existing FL methods to real-world datasets is the data heterogeneity in the data distribution between participating clients [4, 5].

There have been a plethora of works proposing solutions to FL under Non-IID data in recent years. They can be categorized into four groups: 1) alleviating non-guaranteed and weight divergence [6, 7, 8, 9, 10, 11], where the local objectives of the clients is modified such that the local model is consistent with the global model to some extent; 2) Modifying the aggregation scheme at the server

Submitted to 36th Conference on Neural Information Processing Systems (NeurIPS 2022). Do not distribute.

side [12, 13, 14, 15, 10]; 3) data sharing [16, 17, 18, 19], where the server shares a small subset of an auxiliary dataset with the clients to help construct a more balanced and IID data distribution on the client; personalized federated learning [20, 21, 4, 22, 23, 24, 25], take the advantage of a certain level of personalization in training the individual clients models rather than training a single global model.

*After aggregating results across 45 papers addressing data heterogeneity in FL, we believe the right approach to tackling data heterogeneity is a highly non-trivial question on which the FL community has barely scratched the surface. In particular, the challenge comes from various facets, including but not limited to:*

1. The FL community lacks a true notion of data heterogeneity without which the provided solutions may not be effective.

2. The community suffers from a lack of standardized benchmarks on which all proposed algorithms be compared. There has been several heterogeneity benchmarks including Non-IID (2), Non-IID (1), Dir(.), and rotated datasets [23, 26, 27] and even more to say. Firstly, not all proposed algorithms compared their method with others on a unique Non-IID setup and secondly, we will show in Section 2.4, and B.3 that the above-mentioned Non-IID setups are more like IID.

3. It is still not well understood in the community whether and under which conditions clients benefit from collaboration under data heterogeneity setting.

To address this situation, we identify issues with current practices, suggest concrete remedies by defining a new notion of data heterogeneity framework in FL which further facilitates standardized evaluations and comparison of methods. Through extensive studies, we have several key findings:

- It is not clear how the existing FL algorithms tackle the data heterogeneity while they lack systematically understanding the data heterogeneity.

- Many of the prior works have emphasized that the statistical data heterogeneity in FL has harmful effects and can lead to poor convergence [20, 4, 28] which necessitate personalization [28, 7]. In contrast, we found that the current data partitioning strategies may not necessarily bring significant challenges in learning accuracy of FL algorithms. Refer to Sections 2.3, 2.4, B.3, and B.4.

- Under the new notion of heterogeneity that will be proposed herein, data heterogeneity can have detrimental effects such that for some of the clients it is not justified to participate in federation. Refer to Table 4 and Section 5.

- None of the existing state-of-the-art (SOTA) FL algorithms beats the others according to the new notion of data heterogeneity that will be presented in this paper. Refer to Table 4.

## 1.1 Current Non-IID Setups

Current practices [3, 14, 27, 20, 16, 21, 23, 26] have very rigid data partitioning strategies among parties, which are hardly representative and thorough. In the experiments of existing studies, data heterogeneity have been simply modeled as Non-IID label skew $(20\%)$, Non-IID label skew $(30\%)$, and Dir$(\alpha)$, or has been generated by augmenting the datasets using rotation [23].

For Non-IID label skew $(20\%)$ and $(30\%)$, $20\%$ and $30\%$ of the total classes in a dataset is randomly assigned to each client, respectively [21]. Then, the samples of each class is randomly and equally partitioned and distributed amongst the clients who own that particular class. For Non-IID Dir$(\alpha)$, we get random samples for class $c$ from Dirichlet distribution according to $p_c \sim \text{Dir}(\alpha)$ and give each client $j$ random samples of class $c$ according to $p_{c,j}$ proportion. In this setup, heterogeneity can be controlled by the parameter $\alpha$ of Dirichlet distribution [26, 13, 16, 5, 27].

These partitioning strategies cannot design a real and comprehensive view of Non-IID data distribution. *As will be delineated later on, the above-mentioned Non-IID partitions that the prior algorithms has been tested on is more like an IID partition because the data distributions of clients are the sub-distributions of a unique dataset such as CIFAR-10. Besides, all clients have a high percentage of label overlap which mimics IID.* That's why it is a common belief that users can benefit from heterogeneity by federation. While in practice, the union of the clients data may not be a only one dataset. For instance, in mobile phones, or recommendation systems, clients may own very different categories of images like animals, celebrities, nature, paintings; advertisement platforms might need to send different categories of ad to the customers. Therefore, due to the small intra-class distance (similarity between distribution of the classes) in the used benchmark datasets, all baselines benefited

highly from federation. This is the reason that heterogeneity has never been a challenge. More discussion on this will be provided in the rest of paper. We break the barrier of experiments on Non-IID data distribution challenges in FL by proposing a new look into data heterogeneity. This approach addresses a broad range of data heterogeneity issues beyond simpler forms of Non-IIDness like label skews. Here we formally introduce our proposed paradigm, where the goal is to define a new notion of data heterogeneity and suggest standard and real Non-IID setups. We hope that this notion, along with introduced setups, be an standard and inspire the federated learning community.

## 2 Overview

### 2.1 Preliminaries

**Principal angles between two subspaces.** Let $\mathcal{U} = \text{span}\{\mathbf{u}_1, ..., \mathbf{u}_p\}$ and $\mathcal{W} = \text{span}\{\mathbf{w}_1, ..., \mathbf{w}_q\}$ be $p$ and $q$-dimensional subspaces of $\mathbf{R^n}$ where $\{\mathbf{u}_1, ..., \mathbf{u}_p\}$ and $\{\mathbf{w}_1, ..., \mathbf{w}_q\}$ are orthonormal, with $1 \leq p \leq q$. There exists a sequence of $p$ angles $0 \leq \Theta_1 \leq \Theta_2 \leq ... \leq \Theta_p \leq \pi/2$ called the principal angles. The sequence of $p$ principal angle between them is defined as

$$\Theta(\mathcal{U}, \mathcal{W}) = \min_{\mathbf{u} \in \mathcal{U}, \mathbf{w} \in \mathcal{W}} \left( \arccos \left( \frac{|\mathbf{u}^T \mathbf{w}|}{\|\mathbf{u}\| \|\mathbf{w}\|} \right) | \mathbf{u} \in \mathcal{U}, \mathbf{w} \in \mathcal{W}, \mathbf{u} \perp \mathbf{u}_j, \mathbf{w} \perp \mathbf{w}_j \right) \quad (1)$$

where $\|.\|$ is the induced norm. The smallest principal angle is $\Theta_1(\mathbf{u}_1, \mathbf{w}_1)$ where the vectors $\mathbf{u}_1$ and $\mathbf{w}_1$ are the corresponding principal vectors. The rest of Preliminaries appear in Appendix A.

### 2.2 Methodology

In our approach the data heterogeneity/homogeneity should be identified by analyzing the principal angles between the client data subspaces. More particularly, each client in FL applies a truncated SVD step on its own local data in a single-shot manner to derive a small set of principal vectors, which form the principal bases of the underlying data. These principal bases provide a signature that succinctly captures the main characteristics the underlying distribution. These principal bases efficiently identifies distribution heterogeneity/homogeneity among clients by comparing the principal angles between the client data subspaces spanned by the provided principal vectors. The greater the difference in data heterogeneity between two clients, the more orthogonal their subspaces.

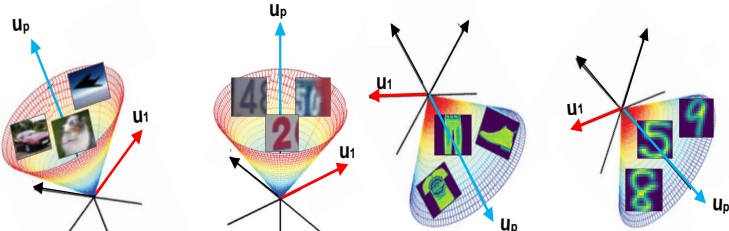

**Figure 1:** There must be a translation protocol enabling the server to understand similarity and dissimilarity in the distribution of data across clients without sharing data. These 2D figures intuitively demonstrate how the principal angle between the client data subspaces captures the statistical heterogeneity. In particular, it shows the subspaces spanned by the $\mathbf{U}_p$s of four different datasets (left to right: CIFAR-10, SVHN, FMNIST, and USPS). As can be seen the principal angle between the corresponding $\mathbf{u}$ vectors of CIFAR-10 and SVHN is smaller than that of CIFAR-10 and USPS. Table 2.2 shows the exact principal angles between every pairs of these subspaces.

To measure the statistical heterogeneity among different users' domains, in this paper, we leverage the angle between clients data subspaces spanned by the most significant left singular vectors of clients data. To begin, we introduce the data heterogeneity via the new notion presented in this paper and then we will generate Non-IID data partitioning across the clients using the proposed method. For a dataset, $\mathbf{D}$, we put the data of each class $C_i$ in the columns of its corresponding matrix $Q_i$. We then, perform truncated SVD on $Q_i$ and obtain $\mathbf{U}_p^i = [\mathbf{u}_1, \mathbf{u}_2, ..., \mathbf{u}_p]$, $(p \ll \text{rank}(\mathbf{D}_k))$. These $\mathbf{U}_p$s span the class subspace and provide a useful signature for distinguishing the underlying distributions of each class in $\mathbf{D}$ because these principal bases characterize the main trends in the data of clients (like eigenfaces). Then according to the principal angle in between of the class subspaces, we understand how similar/dissimilar two classes are based on which we can generate Non-IID data partitioning. *The more orthogonal two subspaces are the more heterogeneous the data of two classes will be.*

Having the data *signature* of all classes of dataset $\mathbf{D}$, in hand, we can obtain a proximity matrix $\mathbf{A}$ either as in Eq. (2) whose entries are the smallest principle angle between the pairs of $\mathbf{U}_p^i$ or as in Eq. (3) whose entries are the summation over the angle in between of the corresponding $\mathbf{u}$ vectors (in identical order) in each pairs of $\mathbf{U}_p^i$ (where $\mathbf{tr}(.)$ is the trace operator).

$$\mathbf{A}_{i,j} = \Theta_1\left(\mathbf{U}_p^i, \mathbf{U}_p^j\right), \ \ i,j = 1,...,|\mathcal{C}| \tag{2}$$

$$\mathbf{A}_{i,j} = \mathbf{tr}\left(\arccos\left(\mathbf{U}_p^i * \mathbf{U}_p^j\right)\right), \ \ i,j = 1,...,|\mathcal{C}| \tag{3}$$

where $\mathcal{C}$ is the total number of classes of $\mathbf{D}$. *Either of Eq. 2 and Eq. 3 can be employed in practice, however, theoretically Eq. 3 is more rigorous.* Now, in order to capture the similarity/dissimilarity of different classes of $\mathbf{D}$, we could form disjoint clusters of classes. For forming disjoint clusters, we can perform agglomerative hierarchical clustering [29] on the proximity matrix $\mathbf{A}$. The best number of clusters can easily be determined just by analyzing the proximity matrix. Each cluster contain classes which are roughly identically distributed.

**Figure 2:** UMAP visualization of four different datasets including CIFAR-10 (orange), SVHN (blue), FMNIST (green), USPS (red).

| Dataset | CIFAR-10 | SVHN | FMNIST | USPS |
|---|---|---|---|---|
| CIFAR-10 | 0 (0) | 6.13 (12.3) | 45.79 (91.6) | 66.26 (132.5) |
| SVHN | 6.13 (12.3) | 0 (0) | 43.42 (86.8) | 64.86 (129.7) |
| FMNIST | 45.79 (91.6) | 43.42 (86.8) | 0 (0) | 43.36 (86.7) |
| USPS | 66.26 (132.5) | 64.86 (129.7) | 43.36 (86.7) | 0 (0) |

**Table 1:** An example showing how distribution similarities among different datasets can be accurately estimated by the principal angles between the datasets subspaces. This table shows the proximity matrix of four datasets whose UMAP visualization was shown in Fig. 3 (c). Entries are $x(y)$, where $x$ and $y$ are obtained from Eq. 2, and Eq. 3, respectively. We let of $p$ in $\mathbf{U}_p$ be 2.

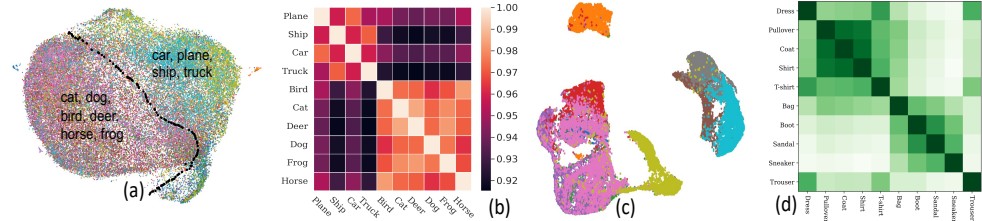

**Figure 3:** The main goal of this figure is to understand the cluster structure of different datasets based on which the Non-IID data partitioning of different datasets can be suggested. (a) depicts the UMAP visualization of CIFAR-10 classes. As can be seen, CIFAR-10 naturally has two super clusters, namely animals (cat, dog, bird, deer, horse, frog) and vehicles (car, plane, ship, truck), which are shown in the purple and green regions, respectively. This means that within each super cluster, the distance between the distribution of the classes is small. While the distance between the distributions of the two super clusters are quite huge. Since the union of clients data is CIFAR-10, two cluster is enough to handle the Non-IIDness across clients. (b) We obtained the proximity matrix A as in Eq. 2 and sketched it. The entries of A are the smallest principle angle between all pairs of classes of CIFAR-10. This concurs with (a) showing the cluster structure of CIFAR-10. (c) The data of FMNIST is naturally clustered into three clusters. The structure of the three clusters is also perfectly suggested by our new proposed notion of heterogeneity for this dataset. (d) We did the same thing as in (b) for FMNIST as well and sketched the matrix.

Before providing more details about the application of the proposed method in defining new Non-IID partition, we elucidate how the proposed method perfectly distinguishes different datasets based on their hidden data distribution by inspecting the angle between their data subspaces spanned by their first $p$ left singular vectors. For a visual illustration of the result, we refer to Fig. 1, Fig. 2, and Table 2.2, where the similarity and dissimilarity of the four different datasets have been evaluated by the proposed measure in Eq. 2, and Eq. 3. Table 2.2 further confirmed with UMAP [30] visualization in Fig. 2. As can be seen from Fig. 2 CIFAR-10 is more similar to SVHN than USPS which reflected in a smaller principal angle between the subspaces of CIFAR-10 and SVHN than that of CIFAR-10 and USPS. It is noteworthy that the smallest principal angle between each pairs of classes is different as well. For instance, on FMNIST, by setting $p$ of $\mathbf{U}_p$ to 3, the smallest principal angle between Trouser and Dress classes is $22.47°$ while that of Trouser and Bag clasees is $51.7°$ which stems from more similarity between the distribution of (Trouser, Dress) compared to that of (Trouser, Bag).

Making use of the the proposed approach for measuring the similarity/dissimilarity of clients data and invoking the proximity matrix of all clients data under the current Non-IID portioning method, we will uncover that the existing data partition strategies represent more like an IID partitioning or at most a light Non-IID that may not necessarily be considered as a challenge. Therefore, this motivates us to propose a new partitioning method using our metric, that leads to the following section.

## 2.3 New Non-IID Partitioning Method

Now we are in position to define a new Non-IID partitioning. We say that *a federated network is heterogeneous if each client owns data only from one of super clusters of a dataset.* In particular, we partition all the training samples in each super cluster into shards of $n$ examples and randomly assign two shards to each client only from one of the super clusters. Such Non-IID (2) partitioning is reasonable to expect in heterogeneous federated networks, due to the drastic disparities in the distribution of each client data. In contrast, the data partitioning proposed in previous papers [16, 4], may assign data from all the clusters.

Consider an example. In Table 2, we show the average final top-1 test accuracy of all clients on CIFAR-10 for FedAvg under IID (2), the conventional Non-IID (C-NIID) label skew (2), and our proposed Non-IID partitioning which we name it as super cluster based Non-IID (SC-NIID). As can be seen from Table 2, the C-NIID data partitioning yield accuracy results close to IID data partitioning while our proposed SC-NIID partitioning accuracy results are far less than that of IID. This shows that the C-NIID data partitioning is not a severe Non-IID and it rather tends to be similar to IID. We will compare the performance of various global and personalized baselines under these Non-IID partitioning methods later on Experiment Section along with some intuitions and remarks.

**Table 2:** Test accuracy comparison on CIFAR-10 across different data partitioning methods. For each partitioning method, the average of final local test accuracy over all clients is reported. We run the FedAVg baseline for each partition 3 times for 100 communication rounds with 10 local epochs and a local batch size of 10.

| Dataset | IID (2) | C-NIID (2) | SC-NIID (2) |
|---|---|---|---|
| CIFAR-10 | $88.15 \pm 0.47$ | $83.63 \pm 1.27$ | $75.53 \pm 3.83$ |

We provide another empirical example to show that our proposed method can effectively produce a more challenging Non-IID data partitioning by leveraging the principal angle between of the data subspaces spanned by the first $p$ significant left singular vectors of the data. In particular, as shown in Table 3, we compare the average distance between the data of each partitioning method including IID, C-NIID, and our proposed SC-NIID. We employ the well-known distance measures including Earth Mover's Distance (EMD) [31], Centered Kernel Alignment (CKA) [32], and our proposed method as in Eq. 2, and Eq. 3 in inspecting the distance between the data of each Non-IID partitioning method. In all of these measures, the smaller the entry is the more IID the data of the partition is. As can be seen, the entries of C-NIID is smaller than that of SC-NIID and are closer to that of IID. Table 2, and 3 together reveal that firstly the C-NIID is more like an IID partitioning or at least it is not a challenging and severe Non-IID and secondly, they demonstrate that our proposed method as in Eq. 2 and Eq. 3 can accurately capture the similarity/dissimilarity between two distributions and its results are consistent with that of the well-known distance measures.

The UMAP [30] visualization in Fig. 3(a) confirms that the images of CIFAR-10 can naturally be clustered into two super clusters, i.e., cluster of animals (cat, dog, deer, frog, horse, bird) and cluster of vehicles (airplane, automobile, ship and truck). This shows that the two clusters is the best case for training the local models on partitions of CIFAR-10 dataset in a Non-IID fashion. Fig 3(b) also

**Table 3:** The average distance, which is evaluated by the well-known distance measures, between all 100 participant clients data under different partitioning methods.

| Measure | IID | C-NIID (2) | SC-NIID (2) |
|---|---|---|---|
| EMD | 0.042 | 0.17 | 0.29 |
| Eq. 2 | 0.072 | 0.202 | 0.274 |
| Eq. 3 | 0.16 | 0.28 | 0.338 |
| CKA | 0.94 | 0.889 | 0.825 |

depicts the proximity matrix of CIFAR-10 dataset, whose entries are the principal angle between the subspace of every pairs of 10 classes (labels). This further confirms that our proposed notion perfectly captures the heterogeneity, thereby finding the best number of super clusters in each dataset. In particular, our experiments demonstrate that the clients that have the sub-classes of these two big classes have common features and can improve the performance of other clients that own sub-classes of the same big class if they be assigned to the same cluster. A similar discussion can be made about other datasets. In contrast, in almost all of the prior works [16, 21, 7, 8, 3], the data samples with the same label are divided into subsets and each client is only assigned two subsets with different labels which produces a very light Non-IID partition as discussed above. Similar settings has been used where each party only has data samples with a single label [3].

Another method that has been adopted in the literature to simulate Non-IID label skew is allocating a proportion of the data of each label/class according to Dirichlet distribution ($\text{Dir}(\alpha)$). In particular, random samples for class $c$ from Dirichlet distribution according to $p_c \sim \text{Dir}(\alpha)$ is selected from the *whole dataset* and to each client $j$ random samples of class $c$ according to $p_{c,j}$ proportion is assigned. While $\text{Dir}(\alpha)$ label skew can simulate label imbalance in the network, it fails to simulate a real data heterogeneity across clients because the assigned samples are randomly selected from the whole

208 dataset and they are not necessarily from only one of the super clusters of that dataset. We rather
209 suggest random samples for each class be selected from each super-cluster on a dataset according to
210 Dirichlet distribution. This will provide a more severe and challenging Non-IIDness. We postpone
211 the experiments on this new $\text{Dir}(\alpha)$ Non-IID setup to sections B.1, B.2, and B.3 in the Appendix.

212 It is noteworthy that the number of formed super clusters in each dataset can be controlled by
213 the distance threshold (linkage) which is a hyperparameter in hierarchical clustering. The smaller
214 the clustering threshold the larger number of super clusters will be formed and in turn the more
215 heterogeneous the partitioning will be. Therefore, the level of data heterogeneity across clients can
216 be easily controlled by the clustering threshold.

### 2.4 A Closer Look at FL Under Heterogeneity

218 To understand which of the Non-IID setups is a better benchmark to be considered in heterogeneous
219 FL scenarios, we perform an experimental study on heterogeneous local models. We choose CIFAR-
220 10 with 10 clients and LeNet-5 as convolutional neural network with 5 layers. We then train the model
221 two times independently with the same random seeds with FedAvg, where in the first training we
222 partition the data according to the C-NIID (2) and in the second time we partition the data according
223 to our new notion of Non-IID-ness i.e., SC-NIID (2). We train both cases for 100 rounds and each
224 client optimizes for 10 local epochs at each round. For each layer in the models, we use CKA [32]
225 and our proposed measure in Eq. 2 to evaluate the similarity of the output features between two local
226 models, given the same input testing samples for each of the cases independently. CKA outputs a
227 similarity score between 0 (not similar at all) and 1 (identical).

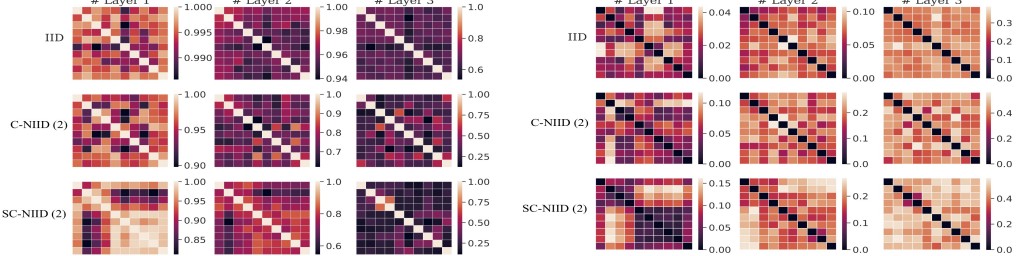

**Figure 4:** Similarities of the outputted feature representation of three different layers of different partitions obtained by CKA (left) and by our proposed measure as in Eq. 2 (right) when trained on CIFAR-10. This plot is sketched once federation finished.

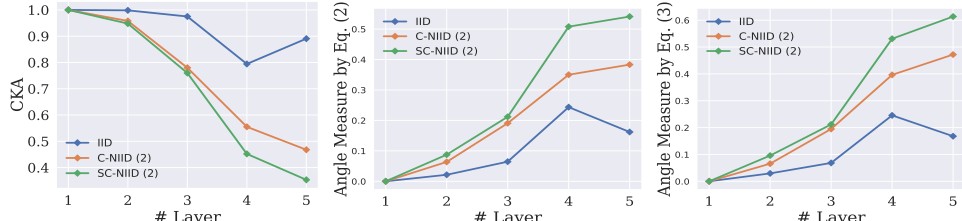

**Figure 5:** The means of the similarities of different layers in different local models obtained by CKA (left) and by our proposed measure as in Eq. 2 (middle) and Eq. 3 (right).

228 We first show the pairwise CKA features similarity of the first three layers of LeNet-5 across local
229 models in Fig. 4. Interestingly, as can be seen, we find that features outputted from IID data and our
230 proposed Non-IID setup show lower CKA similarity compared to that of the C-NIID. It indicates that,
231 our proposed benchmark provide a more severe heterogeneity across different clients. By averaging
232 the pairwise CKA features similarity in Fig. 4, we can obtain a single value to approximately represent
233 the similarity of the feature outputs by each layer across different clients for IID, C-NIID and our
234 SC-NIID setups. We demonstrated the approximated layer-wise features similarity in Fig. 5. These
235 results witness that the models trained on our proposed Non-IID setup consistently produced features
236 across clients for all layers which are less similar to IID in comparison to that of C-NIID.

## 3 Conclusion

238 We proposed a new notion and framework for Non-IID partitioning in FL setups. A dataset is first
239 divided into several super clusters by analyzing the principal angles between subspaces of different
240 classes. To distribute heterogeneous data to all clients, the training data in each super cluster is
241 partitioned to different shards. Each client is assigned a certain number of shards from only one of
242 the super clusters. The proposed method addresses a broad range of data heterogeneity issues beyond
243 simpler forms of Non-IIDness like label skews.

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

# Appendix

The Appendix is organized as follows. Section A provides additional preliminaries; Section B elaborate upon more experiments to delineate the performance of the proposed approach; finally, Section C contains implementation details.

## A    Preliminaries

This section complements section 2.1 of the main paper.

**Truncated Singular Value Decomposition (SVD).** Truncated SVD of a real $m \times n$ matrix $\mathbf{M}$ is a factorization of the form $\tilde{\mathbf{M}} = \mathbf{U}_p \mathbf{\Sigma}_p \mathbf{V}_p^T$ where $\mathbf{U}_p = [\mathbf{u_1}, \mathbf{u_2}, ..., \mathbf{u_p}]$ is an $m \times p$ orthonormal matrix, $\mathbf{\Sigma_p}$ is a $p \times p$ rectangular diagonal matrix with non-negative real numbers on the main diagonal, and $\mathbf{V}_p = [\mathbf{v}_1, \mathbf{v}_2, ..., \mathbf{v}_p]$ is a $p \times n$ orthonormal matrix, where $\mathbf{u}_i \in \mathbf{U}_p$ and $\mathbf{v}_i \in \mathbf{V}_p$ are the left and right singular vectors, respectively.

**Hierarchical clustering.** When forming disjoint clusters where the number of clusters is not known in advance, hierarchical clustering (HC) [29] is of interest. Agglomerative HC is one of the well-known clustering techniques in machine learning that takes a proximity (adjacency) matrix as input and groups similar objects into clusters. HC starts by treating each client as a separate cluster. At each step of the clustering, the pairwise $L_2$ (Euclidean) distance between all clusters is computed to identify their similarity. The two clusters that are most closest ones are merged. This iterative process continues till all the clusters are merged into one. And finally, a distance threshold is defined to determine when to stop merging clusters. In this paper the distance threshold is called clustering threshold.

## B    Experiments

We perform an extensive empirical analysis using a standard image classification task for multiple popular federated learning datasets and various statistical heterogeneity setups.

### B.1    Experimental Setup

**Datasets and Models**. We use image classification task and 3 popular datasets, i.e., CIFAR-10 [33], CIFAR-100 [33], STL-10 [34] to employ our novel partitioning method. For all experiments, we consider LeNet-5 [35] architecture for CIFAR-10 dataset and ResNet-9 [36] architecture for CIFAR-100, and STL-10 datasets. Details of the architectures can be found in subsection C.

**Baselines and Implementation**. To assess the performance of our novel Non-IID partitioning method against the conventional partitioning, we compare the results over a set of baselines. For SOTA personalized FL methods, the baselines include LG-FedAvg [27], Per-FedAvg [4], Clustered-FL (CFL) [24], and IFCA [23]. Besides, we compare with FedAvg$^+$ [3], FedProx$^+$ [8] FedNova$^+$ [14], and SCAFFOLD$^+$ [9]. It is noteworthy that the superscript $+$ sign on global baselines means that these baselines has been fine-tuned by the clients and thus are considered personalized ones. We report the average results performance over three independent trials.

**C-NIID ($\varrho\%$) Label Skew and SC-NIID ($\varrho\%$).** In this setting, the conventional method first randomly assigns $\varrho\%$ of the total available labels of a dataset to each client and then randomly distribute the samples of each label amongst clients own those labels as in [37]. In our SC-NIID method, we first form the super clusters according to our proposed method explained in Section 2. We then randomly assign all clients to only one of the formed super clusters. The number of assigned clients to each super cluster is proportional to the size (number of samples that cluster contains) of the super cluster which means that if the size of a super cluster is bigger, more number of clients are assigned to that particular super cluster. Next, the total samples of each super cluster is divided into shards and each client pick $\varrho\%$ of the total labels contained in the super cluster which the client belongs to.

**Table 4:** Evaluating different personalized FL methods under different data partitions. We evaluate on ResNet-9 with CIFAR-100 and STL-10 as well as LeNet-5 on CIFAR-10. For each communication round, a fraction 10%, 30%, 10% of the total 100 clients are randomly selected. We set local epoch and batch size to 10.

| Dataset | Algorithm | C-NIID(2) | SC-NIID(2) | C-Dir(0.5) | SC-Dir(0.5) |
|---|---|---|---|---|---|
| CIFAR-10 | SOLO | $83.62 \pm 0.72$ | $75.68 \pm 0.47$ | $48.45 \pm 0.57$ | $43.44 \pm 0.43$ |
| | FedAvg+ | $83.46 \pm 1.15$ | $77.02 \pm 0.73$ | $53.31 \pm 0.85$ | $43.43 \pm 1.81$ |
| | FedProx+ | $85.01 \pm 0.59$ | $76.54 \pm 1.15$ | $52.69 \pm 0.60$ | $44.61 \pm 1.43$ |
| | FedNova+ | $83.99 \pm 0.68$ | $77.36 \pm 0.46$ | $53.21 \pm 0.82$ | $46.09 \pm 0.33$ |
| | Scafold+ | $82.69 \pm 2.93$ | $78.88 \pm 0.44$ | $41.55 \pm 5.82$ | $16.45 \pm 2.71$ |
| | LG | $83.18 \pm 0.46$ | $76.40 \pm 0.46$ | $31.35 \pm 7.42$ | $41.36 \pm 0.75$ |
| | PerFedAvg | $83.60 \pm 0.48$ | $75.70 \pm 0.74$ | $54.90 \pm 0.25$ | $42.63 \pm 1.08$ |
| | IFCA | $86.59 \pm 1.16$ | $80.49 \pm 0.86$ | $57.78 \pm 1.14$ | $51.54 \pm 0.98$ |
| CIFAR-100 | SOLO | $76.71 \pm 1.12$ | $73.16 \pm 0.17$ | $45.24 \pm 1.74$ | $40.43 \pm 0.85$ |
| | FedAvg+ | $87.99 \pm 0.97$ | $81.55 \pm 0.59$ | $66.15 \pm 2.79$ | $53.41 \pm 1.16$ |
| | FedProx+ | $87.68 \pm 0.82$ | $80.70 \pm 0.81$ | $67.67 \pm 0.98$ | $53.86 \pm 0.25$ |
| | FedNova+ | $87.22 \pm 0.45$ | $80.75 \pm 1.28$ | $63.15 \pm 2.32$ | $53.77 \pm 0.52$ |
| | Scafold+ | $55.97 \pm 13.29$ | $15.61 \pm 9.85$ | $39.04 \pm 23.73$ | $11.43 \pm 3.90$ |
| | LG | $78.27 \pm 1.31$ | $72.87 \pm 0.80$ | $44.43 \pm 1.40$ | $39.44 \pm 0.90$ |
| | PerFedAvg | $77.47 \pm 1.30$ | $58.93 \pm 0.90$ | $58.02 \pm 2.38$ | $37.96 \pm 1.24$ |
| | IFCA | $88.06 \pm 0.19$ | $82.23 \pm 0.73$ | $69.89 \pm 1.64$ | $55.66 \pm 0.86$ |
| STL-10 | SOLO | $78.14 \pm 2.27$ | $69.88 \pm 0.62$ | $51.12 \pm 1.16$ | $42.17 \pm 0.68$ |
| | FedAvg+ | $85.67 \pm 2.23$ | $77.23 \pm 1.45$ | $56.80 \pm 5.64$ | $49.69 \pm 0.66$ |
| | FedProx+ | $86.83 \pm 1.83$ | $83.03 \pm 1.40$ | $64.97 \pm 5.86$ | $56.35 \pm 1.85$ |
| | FedNova+ | $88.45 \pm 0.27$ | $83.18 \pm 2.28$ | $60.00 \pm 6.77$ | $52.84 \pm 1.03$ |
| | Scafold+ | $31.74 \pm 1.80$ | $27.71 \pm 4.12$ | $50.31 \pm 2.90$ | $31.28 \pm 7.52$ |
| | LG | $84.42 \pm 0.77$ | $75.52 \pm 0.91$ | $52.56 \pm 2.94$ | $46.82 \pm 1.47$ |
| | PerFedAvg | $52.91 \pm 2.12$ | $51.64 \pm 0.57$ | $30.10 \pm 2.74$ | $31.80 \pm 3.14$ |
| | IFCA | $88.99 \pm 0.45$ | $81.13 \pm 0.46$ | $67.99 \pm 1.66$ | $60.73 \pm 1.27$ |

**Conventional Dir (C-Dir) and SC-Dir.** In this setting, the conventional method distributes the training data between the clients based on the Dirichlet distribution. In particular, for $N$ clients data it samples $N$ random numbers $\mathbf{p}_i \sim Dir_N(\alpha)$ from $Dir(\alpha)$ distribution [1] and allocates the $\mathbf{p}_{i,j}$ proportion of the training data of class $i$ to client $j$ as in [37]. In our proposed SC-Dir(.) method, we again constitute the super clusters with the help of Eq. 2 or Eq. 3 as explained in Section 2. We then assign certain number of clients randomly to only one of these super clusters depending upon the size of each super cluster. We then let the data within each super cluster be assigned according to the Dir(.) distribution as in C-Dir(.) to the clients that belong to each cluster.

## B.2 Comparing the Performance of SOTA Baselines on the Conventional and Newly Proposed Non-IID Method

We conduct experiments to compare the above four Non-IID partitioning method i.e, C-NIID (2), C-Dir(.), SC-NIID (2), and SC-Dir(.) methods on CIFAR-10, CIFAR-100, and STL-10 datasets and present the results in Table 4. It can be observed that all SOTA baselines present a great performance drop when the clients data are distributed according to the newly proposed Non-IID setup compared to the conventional Non-IID setups used in prior arts. Based on the results of table 4 and through extensive studies, we have the following key findings: 1) The newly proposed Non-IID partitioning is more challenging compared to the C-NIID. Since effectively addressing data heterogeneity is of paramount concern in federated learning, we suggest that the challenging tasks like SC-NIID ($\varrho\%$), and SC-Dir($\alpha$) should be included in the benchmark for future FL setups. 2) Under the new Non-IID setup none of the existing SOTA FL algorithms outperforms others in all cases. This further indicates the importance of having a more comprehensive Non-IID distribution benchmark.

---

[1]The value of $\alpha$ controls the degree of Non-IID-ness. A big value of $\alpha$ e.g., $\alpha = 100$ mimics identical label distribution (IID), while $\alpha = 0.1$ results in a split, where the vast majority of data on every client are Non-IID.

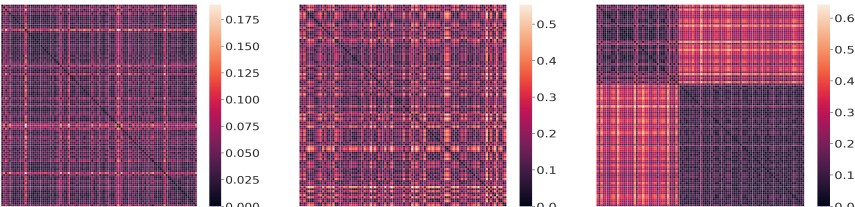

**Figure 6:** We obtained a proximity matrix of $100 \times 100$ dimension which corresponds to 100 clients' data distribution by the EMD measure for three different data partitioning method, i.e., IID (left), C-NIID (middle), and SC-NIID (right). This concurs with Fig. 3 showing that CIFAR-10 naturally have two Non-IID clusters as well with Fig. 7, and Fig. 8.

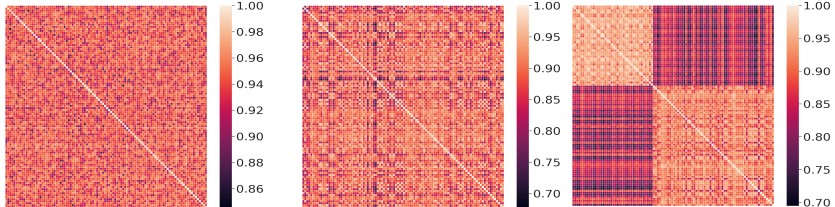

**Figure 7:** We obtained a proximity matrix of $100 \times 100$ dimension which corresponds to 100 clients' data distribution by the CKA measure for three different data partitioning method, i.e., IID (left), C-NIID (middle), and SC-NIID (right). This concurs with Fig. 3 showing that CIFAR-10 naturally have two Non-IID clusters as well with Fig. 6, and Fig. 8

### B.3    Comparing the Level of Data Heterogeneity of the C-NIID and SC-NIID

Data heterogeneity will impact the convergence of a federated model and hurts its performance. This necessitate a deeper investigation of data heterogeneity which has been missing. Herein we provide some visualizations to facilitate understanding the heterogeneity level of the setup that has been used as an standard one in the prior works. To this end, we distribute CIFAR-10 according to three data partitioning method, i.e., IID, C-NIID, and SC-NIID among the clients and then we leverage three distribution similarity/dissimilarity measures namely EMD [31], CKA [32], and our proposed method as in Eq. 2, to monitor the significance of data heterogeneity across 100 participant clients in the federation. Figures 6–8 visualize the proximity matrix whose entries are the distance between clients data computed by EMD, CKA, and our proposed method, respectively. These figures further confirm that the C-NIID partitioning is more like IID. This is while as can be seen in these three figures, our newly proposed SC-NIID distribute the data across all clients which is very different from IID. This corresponds to the fact that our method takes the intrinsic structure of the dataset into account and from a certain number of Non-IID super clusters according to the entries of the proximity matrix and then distribute the data among the clients from only one of these Non-IID super clusters. According to what we discussed, it is therefore natural to ask the following questions: *How do prior FL approaches tackle the newly proposed Non-IID partitioning?*

**Remark-1.** It is not clear how the the prior personalized FL algorithms which has been proposed to alleviate the statistical data heterogeneity should be extended to tackle the new Non-IID setup. For



**Figure 8:** We obtained a proximity matrix of $100 \times 100$ dimension which corresponds to 100 clients' data distribution by our own proposed measure as in Eq. 2 for three different data partitioning method, i.e., IID (left), C-NIID (middle), and SC-NIID (right). This concurs with Fig. 3 showing that CIFAR-10 naturally have two Non-IID clusters as well with Fig. 6, and Fig. 7

instance, as mentioned in [26], the proposed regularization method is effective only for light data heterogeneity but would not be beneficial or even lead to drop in the performance with the increase of the heterogeneity. This could be true about many of the prior arts as also confirmed by the results reported in table 4. This further emphasizes that the performance of the prior arts should be evaluated under severe and challenging Non-IID setups to judge their efficacy.

**Remark-2.** While many works in the literature have highlighted detrimental impacts of statistical data heterogeneity, showing that heterogeneity can lead to poor performance in federated optimization which necessitates novel forms of personalization, some works [28] discussed distinct benefits of data heterogeneity in their FL analyses. In contrast to this work [28], we believe to judge whether the impact of the corresponding heterogeneity issue is positive or negative, we should first provide a true notion of data heterogeneity as well as standard Non-IID benchmarks without which it will be hard to attest measurable benefits of heterogeneity.

**Remark-3.** In Table 4 we provided the results of SOLO training as well. Considering SC-Dir(0.5) on CIFAR-10, as is evident from Table 4, except for a few baselines, the accuracy results of other baselines are worse than SOLO training which means that it is not justified for clients to participate in federation under the newly proposed Non-IID setting. Because the clients not only do not gain from federation but federation also has degraded their performance. This phenomenon can be explained by the fact that depending upon the dataset the proposed Non-IID partitioning distributes highly heterogeneous data to clients under which federation is not beneficial. This is while considering C-Dir(0.5) most of the baselines benefited from federation and yielded better results compared to SOLO training. The same behavior can be seen on other datasets and on SC-NIID(2).

### B.4 Under What Heterogeneity Environment Clients Benefit from Collaboration?

In this section, by an empirical pseudo example we demonstrate that the widely used Non-IID setup, i.e., C-NIID in the literature is not a real Non-IID partition. In doing so, we sample two clients named as C1, and C2 and manually assign two labels to each as in Table 5 and let them to do federation with vanilla FedAvg [3] for 20 communication rounds. When C1 owns labels "ship+truck", and C2 owns labels "plane+car" as in row 4, according to the common belief in the FL community we should have expected that due to the existence of C-NIIDness the average final accuracy be worse than SOLO[2] training as in row 1. But surprisingly it is not the case and even though these clients have no label overlap, their performance improved through federation compared to SOLO. In contrast, as can be seen from row 6, even though C1 and C2 have $50\%$ labels overlap ($50\%$ distribution similarity according to C-NIID), by taking part in federation, the accuracy of C1 drops compared to SOLO. This is while row 4 with the same condition ($50\%$ labels overlap) improved the results of C1 through federation. This further confirms that C-NIID cannot represent a true view of non-IIDness in FL. These anomalies can be justified by our newly proposed Non-IID partition which is based on the simple label skew but based on the angle in between of the clients' data subspaces. In particular, two clients data are considered heterogeneous if their data are drawn from two different super clusters formed by our approach in Eq. 2 or Eq. 3. The result of C1 in row 4 improved because both of the clients own data from the same super cluster. The results of other rows can be justifies in the same fashion.

The authors hope the reader take the preceding discussion as an example showing that the adopted Non-IID partition in the prior works may not represent a true data heterogeneity setup and also the authors do not claim that the proposed method can justify all anomalies. We rather like to encourage the researchers to designing more comprehensive alternatives to the current Non-IID setups.

### B.5 A New Benchmark for Non-IID FL

As mentioned earlier, existing studies have been evaluated on simple partitioning strategies, i.e., Non-IID label skew ($20\%$) and Non-IID label skew ($30\%$). In data partitioning with $a\%$ label skew, the union of client data will only be one dataset. Focusing on CIFAR-10 and with $20\%$ label skew, most of the 100 clients can have either $50\%$ label overlap or $100\%$. This simulates a partially Non-IID setting and cannot represent a full view of Non-IIDness. Because the data distributions of clients are the sub-distributions of a unique dataset such as CIFAR-10. This is the reason that statistical data heterogeneity has never been a big issue in the proposed personalized FL algorithms.

---

[2]In SOLO baseline the client trains a model lonely on it own local data without taking part in federation.

**Table 5:** A pseudo example illustrating that the adopted heterogeneous setup in the prior works which relies upon the label skew can not represent a true view of Non-IIDness. That's why it has not necessarily had detrimental effect on the clients accuracy performance.

| Row | Case | C1 Accuracy | Status |
|---|---|---|---|
| 1 | ship+truck ; [8,9] | 83.20 | Solo Training |
| 2 | C1: ship+truck, C2:ship+truck; [8,9] | 86.02 | Full overlap (IID) |
| 3 | C1: ship+truck, C2:truck+plane; [9,0] | 84.02 | One overlap on vehicle |
| 4 | C1: ship+truck, C2:plane+car; [0,1,8,9] | 83.36 | No overlap (only vehicle) |
| 6 | C1: ship+truck, C2:truck+bird; [9,2] | 82.43 | One overlap on animal |
| 7 | C1:ship+truck, C2:car+bird; [1,2,8,9] | 81.78 | No overlap (animal+vehicle) |
| 8 | C1: ship+truck, C2:ship+bird; [8,2] | 81.72 | One overlap on animal |
| 9 | C1: ship+truck, C2:cat+dog; [3,5,8,9] | 81.63 | No overlap (only animal) |

**Table 6:** The benefits of personalized SOTA algorithms should be testified when the tasks are severely Non-IID. This table evaluates different FL approaches in the challenging scenario of MIX-4 in terms of test accuracy performance. All approaches have substantial difficulties in handling this scenario with tremendous data heterogeneity. We run each baseline 3 times for 50 communication rounds with 5 local epochs.

| Algorithm | MIX-4 |
|---|---|
| SOLO | $55.08 \pm 0.29$ |
| FedAvg | $63.68 \pm 1.64$ |
| FedProx | $61.86 \pm 3.73$ |
| FedNova | $60.92 \pm 3.60$ |
| Scaffold | $69.26 \pm 0.84$ |
| LG | $58.49 \pm 0.46$ |
| PerFedAvg | $42.60 \pm 0.60$ |
| IFCA | $70.32 \pm 3.57$ |
| CFL | $61.18 \pm 2.63$ |

In order to better assess the potential of the SOTA baselines under a real-world and challenging Non-IID task where the local data of clients have strong statistical heterogeneity, and the data distributions of clients are not the sub-distributions of a unique dataset, we design the following benchmark naming it as MIX-4. We hope Mix-4 become an standard benchmark for comparing different SOTA against each other in the FL community. We assume that each client owns data samples from one of the four datasets, i.e., USPS [38], CIFAR-10, SVHN, and FMNIST. In particular, we distribute CIFAR-10, SVHN, FMNIST, USPS among x, y, z, v clients, respectively (x+y+z+v= total clients) where each client receives a certain number of samples from all classes but only from one of these dataset. This is a very challenging Non-IID task. Under this circumstance, the union of the clients data is not a single dataset and in each round of communications there will be some clients whose data distribution vary drastically. In Table 6, we present the test accuracy of the SOTA baselines on Mix-4. It can be seen from Table 6 that the accuracy performance of all the methods drops significantly compared to the ones reported for C-NIID (the widely used Non-IID setup in prior works) in Table 4. These sorts of realistic assumption has never been adopted in the literature. That's why it is a common belief that all users can benefit from heterogeneity. We hope designing Mix-4 opens up a new avenue to design more standard real-world benchmark for comparing different personalized SOTA in the FL community.

**Table 7:** The formed super clusters for each dataset. The numbers correspond to the labels according to the standard naming of labels in each dataset.

| Dataset | Formed Super Clusters |
|---|---|
| CIFAR-10 | {0,1,8,9}, {2,3,4,5,6,7} |
| CIFAR-100 | {0, 83, 53, 82}, {1, 54, 43, 51, 70, 92, 62}, {23, 69, 30, 95, 67, 73}, {47, 96, 59, 52}, {2, 97, 27, 65, 64, 36, 28, 61, 99, 18, 77, 79, 80, 34, 88, 42, 38, 44, 63, 50, 78, 66, 84 , 8, 39, 55, 72, 93, 91, 3, 4, 29, 31, 7 , 24, 20, 26, 45, 74, 5, 25, 15, 19, 32, 9, 16, 10, 22, 40, 11, 35, 98, 46 , 6, 14, 57, 94, 56, 13, 58, 37, 81, 90, 89, 85, 21, 48, 86, 87, 41, 75, 12, 71, 49, 17, 60, 76, 33, 68} |
| STL-10 | {2, 8, 9}, {0, 1, 7, 3, 4, 5, 6} |

# C Implementation

We have released our implemented code at `https://github.com/anonresearcher1/alg-neuripsw22`. To be consistent, we adapt the official public codebase of Qinbin et al. [37] [3] to implement our proposed method and all the other baselines with the same codebase in PyTorch V. 1.9. We used the public codebase of LG [22][4], Per-FedAvg [4] [5], and IFCA [23] [6] in our implementation. For all the global benchmarks, including FedAvg [3], FedProx [8], FedNova [14], Scaffold [9] we used the official public codebase of Qinbin et al. [37] [3]. It is worth noting that, unlike the original paper and the official implementation of LG[22] [4], for the sake of fair comparison we initialized the models randomly with the same random seed just like all other baselines.

**The formed super clusters on each dataset.** The details of formed super clusters on each dataset is presented in table 7.

## C.1 Implementation Details for MIX-4

We set the number of clients to 100 and distribute CIFAR-10, SVHN, FMNIST, USPS amongst 31, 25, 27, 14 clients such that each client receives 500 samples from all the available classes in the corresponding dataset (50 samples per each class). We further zero-pad FMNIST, and USPS images to make them $32 \times 32$, and repeat them to have 3 channels. This pre-processing for FMNIST, and USPS is required to make the images the same size as CIFAR-10 and SVHN so that we can have a consistent model architecture in this task. Tables 9, and 8 present more details about other hyper-parameter grids used in this experiment. Further, we used LeNet-5 architecture with the details in Table 10, and modified the last layer to have 40 outputs corresponding to the 40 number of total labels (each dataset own 10 classes).

## C.2 Hyper-parameters & Architectures

Tables 10, and 11 show the details of the convolutional neural network used for CIFAR-10, CIFAR-100, and STL-10.

---

[3] `https://github.com/Xtra-Computing/NIID-Bench`
[4] `https://github.com/pliang279/LG-FedAvg`
[5] `https://github.com/CharlieDinh/pFedMe`
[6] `https://github.com/jichan3751/ifca`

**Table 8:** Hyper-parameters used for LG, Per-FedAvg, and IFCA throughout the experiments.

| Method | Hyper-parameters | CIFAR-100 | CIFAR-10 | STL-10 |
|---|---|---|---|---|
| LG | model | ResNet-9 | LeNet-5 | ResNet-9 |
| | learning rate | 0.01 | 0.01 | 0.01 |
| | weight decay | 0 | 0 | 0 |
| | momentum | 0.5 | 0.5 | 0.5 |
| | number of local layers | 7 | 3 | 3 |
| | number of global layers | 2 | 2 | 2 |
| Per-FedAvg | model | ResNet-9 | LeNet-5 | ResNet-9 |
| | learning rate | 0.01 | 0.01 | 0.01 |
| | weight decay | 0 | 0 | 0 |
| | momentum | 0.5 | 0.5 | 0.5 |
| | $\alpha$ | 1e-2 | 1e-2 | 1e-2 |
| | $\beta$ | 1e-3 | 1e-3 | 1e-3 |
| IFCA | model | ResNet-9 | LeNet-5 | ResNet-9 |
| | learning rate | 0.01 | 0.01 | 0.01 |
| | weight decay | 0 | 0 | 0 |
| | momentum | 0.5 | 0.5 | 0.5 |
| | number of clusters | 2 | 2 | 2 |

**Table 9:** The hyper-parameters used for FedAvg+, FedProx+, FedNova+, Scaffold+, and SOLO throughout the experiments

| Method | Hyper-parameters | CIFAR-100 | CIFAR-10 | STL-10 |
|---|---|---|---|---|
| FedAvg+ | model | ResNet-9 | LeNet-5 | ReNet-9 |
| | learning rate | {0.1, 0.01, 0.001} | {0.1, 0.01, 0.001} | {0.1, 0.01, 0.001} |
| | weight decay | 0 | 0 | 0 |
| | momentum | 0.9 | 0.9 | 0.9 |
| FedProx+ | model | ResNet-9 | LeNet-5 | ResNet-9 |
| | learning rate | {0.1, 0.01, 0.001} | {0.1, 0.01, 0.001} | {0.1, 0.01, 0.001} |
| | weight decay | 0 | 0 | 0 |
| | momentum | 0.9 | 0.9 | 0.9 |
| | $\mu$ | {0.01, 0.001} | {0.01, 0.001} | {0.01, 0.001} |
| FedNova+ | model | ResNet-9 | LeNet-5 | ResNet-9 |
| | learning rate | {0.1, 0.01, 0.001} | {0.1, 0.01, 0.001} | {0.1, 0.01, 0.001} |
| | weight decay | 0 | 0 | 0 |
| | momentum | 0.9 | 0.9 | 0.9 |
| Scaffold+ | model | ResNet-9 | LeNet-5 | ResNet-9 |
| | learning rate | {0.1, 0.01, 0.001} | {0.1, 0.01, 0.001} | {0.1, 0.01, 0.001} |
| | weight decay | 0 | 0 | 0 |
| | momentum | 0.9 | 0.9 | 0.9 |
| SOLO | model | ResNet-9 | LeNet-5 | ResNet-9 |
| | learning rate | 0.01 | 0.01 | 0.01 |
| | weight decay | 0 | 0 | 0 |
| | momentum | 0.5 | 0.5 | 0.5 |

**Table 10:** The details of LeNet-5 architecture used for the FMNIST, SVHN, CIFAR-10, and Mix-4 datasets.

| Layer | Details |
|---|---|
| layer 1 | Conv2d(i=3, o=6, k=(5, 5), s=(1, 1))
ReLU()
MaxPool2d(k=(2, 2)) |
| layer 2 | Conv2d(i=6, o=16, k=(5, 5), s=(1, 1))
ReLU()
MaxPool2d(k=(2, 2)) |
| layer 3 | Linear(i=400 (256 for FMNIST), o=120)
ReLU() |
| layer 4 | Linear(i=120, o=84)
ReLU() |
| layer 5 | Linear(i=84, o=10 (100 for CIFAR-100, and 40 for Mix-4)) |

**Table 11:** The details of ResNet-9 architecture used for CIFAR-100, and STL-10 dataset.

| Block | Details | Input |
|---|---|---|
| block 1 | Conv2d(i=3, o=64, k=(3, 3), s=(1, 1))
GroupNorm(g=32, o=64)
ReLU() | image |
| block 2 | Conv2d(i=64, o=128, k=(3, 3), s=(1, 1))
GroupNorm(g=32, o=128)
ReLU()
MaxPool2d(k=(2, 2)) | block 1 |
| block 3 | Conv2d(i=128, o=128, k=(3, 3), s=(1, 1))
GroupNorm(g=32, o=128)
ReLU()
Conv2d(i=128, o=128, k=(3, 3), s=(1, 1))
GroupNorm(g=32, o=128)
ReLU() | block 2 |
| block 4 | Conv2d(i=128, o=256, k=(3, 3), s=(1, 1))
GroupNorm(g=32, o=256)
ReLU()
MaxPool2d(k=(2, 2)) | block 2 +
block 3 |
| block 5 | Conv2d(i=256, o=512, k=(3, 3), s=(1, 1))
GroupNorm(g=32, o=512)
ReLU()
MaxPool2d(k=(2, 2)) | block 4 |
| block 6 | Conv2d(i=512, o=512, k=(3, 3), s=(1, 1))
GroupNorm(g=32, o=512)
ReLU()
Conv2d(i=512, o=512, k=(3, 3), s=(1, 1))
GroupNorm(g=32, o=512)
ReLU() | block 5 |
| classifier | MaxPool2d(k=(4, 4))
Linear(i=512, o=100) | block 4 +
block 5 |

