# OpenReview forum: "Rethinking Data Heterogeneity in Federated Learning: Introducing a New Notion and Standard Benchmarks"
_NeurIPS.cc/2022/Workshop/Federated_Learning — FL-NeurIPS 2022 Poster_

### Official Review · Reviewer_LoX9 · 2022-10-18
**Review of the paper**

## Summary

The paper shows that standard benchmarks used in FL (FMNIST and Cifar) actually have quite a lot of structure in them - their labels can be grouped into a small number of clusters. This is also intuitive - car and trucks are ostensibly much more similar to each other than to cats and dogs. Thus, simply partitioning by label may not be the most challenging way to create a non-iid dataset. Instead, the paper first proposes to perform some spectral clustering of the datapoints and use this clustering to create highly non-iid partitions. They show that FedAvg when trained on such partitions performs worse.

## Review

While this is an interesting paper, I think the ideas investigated here are still premature.

1. The authors seem to be trying to create "adversarial data partitions", even though this is not how they frame it in this work. Explicitly framing this as an adversarial setting may yield more insights and a broader scope of possible techniques.
2. They also suggest that their measure of dissimilarity more accurately captures the difficulty of the problem. However, there isn't sufficient evidence provided for this claim. The authors could create some datasets (either synthetic or natural) with varying levels of difficulty for FedAvg. Then, the various metrics of similarity can be compared against one other using a correlation analysis to see which metric more accurately captures the difficulty. Such an analysis would be very interesting and a valuable contribution.
3. Finally, the paper mostly focuses on FedAvg. While this is understandable given FedAvg's popularity it would be great to see the result on other popular algorithms as well.

---

### Official Review · Reviewer_VBmU · 2022-10-19
**Good effort on data heterogeneity, but the claims appear to be too strong**

## Strengths:
* The paper introduces an interesting way to create more challenging Non-IID settings for FL.
* The proposed benchmark and open-source code could be useful for other research efforts.
## Weaknesses:
* The paper seems to overclaim its contributions on many occasions. For example:
     1. The paper highlights 20% or 30% Non-IID, but it is well-known that these are not the most challenging ones. As the paper mentions, many existing work also uses Dirichlet distribution or extreme Non-IID, which is much more challenging than the 20% or 30% C-NIID.
    2. There are quite a few FL benchmarks for real-world data heterogeneity such as LEAF, FedScale and FedML. These are more realistic than different synthetic partitioning, where the proposed method belongs to.
    3. The paper claims that Dir($\alpha$) fails to simulate real data heterogeneity, but it is unclear why the proposed Non-IID setup is more realistic. It appears that the proposed method is more challenging but not necessarily more realistic.
* The paper could have been stronger if the experiment is done on real-world data partitions.
* It would be good for the paper to explain how the proposed method can be applied to other types of data such as natural language processing.

---

### Official Review · Reviewer_1hiH · 2022-10-19
**Review of rethinking data heterogeneity**

This paper discusses a fundamental issue of data heterogeneity in FL. The authors studied the challenges of data heterogeneity in FL and defined a notion and framework for the non-i.i.d. data partitioning. It shows that the proposed SC-non-i.i.d. method poses more challenges rather the classic ones and is able to capture the similarity/dissimilarity between two distributions, which would be useful for data partitioning. In general, the paper is well written. But there is no any theoretical justification for the method. Multiple experiments verify the idea of the super cluster based non-i.i.d.

---

### Decision · Program_Chairs · 2022-10-20

Accept (Poster)